# Characteristics of the gut microbiota in women with premenstrual symptoms: A cross-sectional study

**Takashi Takeda**[1]*, **Kana Yoshimi**[1], **Sayaka Kai**[1], **Genki Ozawa**[2], **Keiko Yamada**[3], **Keizo Hiramatsu**[4]

**1** Division of Women's Health, Research Institute of Traditional Asian Medicine, Kindai University, Osaka-Sayama, Osaka, Japan, **2** TechnoSuruga Laboratory Co. Ltd., Shizuoka, Japan, **3** Department of Anesthesiology and Pain Medicine, Juntendo University Faculty of Medicine, Tokyo, Japan, **4** Hiramatsu Women's Clinic, Fujiidera, Osaka, Japan

* take@med.kindai.ac.jp

**Data Availability Statement:** All relevant data are available in the paper and Supporting Information files. The 16S rRNA gene sequences from the study participants analysed in this study were

## Abstract

### Purpose

Premenstrual symptoms can negatively impact the quality of life of women through a range of mood, behavioral, and physical symptoms. The association between the microbiota and brain function has been extensively studied. Here, we examined the characteristics of the microbiota in women with premenstrual disorders (PMDs) and the association between premenstrual symptoms and the microbiota.

### Materials and methods

In this single center cross-sectional pilot study, we recruited 27 women reporting premenstrual symptoms and 29 women with no serious premenstrual symptoms. Among them, we further selected 21 women experiencing premenstrual symptoms resulting in interference to their social life (PMDs group) and 22 women with no serious premenstrual symptoms and thereby no interference to their social life (control group). The severity of symptoms was evaluated by a premenstrual symptoms questionnaire (PSQ). Inflammatory markers were analyzed in blood samples, including C reactive protein, soluble CD14, and lipopolysaccharide binding protein. Sequencing of 16S ribosomal ribonucleic acid genes was performed on stool samples.

### Results

Inflammatory markers in blood samples did not differ significantly between the PMDs and control groups. A difference in beta, but not alpha diversity, was detected for the gut microbiotas of the PMDs and control groups. The relative abundance of the *Bacteroidetes* phylum was lower in the PMDs group. At the genus level, the prevalence was decreased for *Butyricicoccus*, *Extibacter*, *Megasphaera*, and *Parabacteroides* and increased for *Anaerotaenia* in the PMDs group, but after false discovery rate correction, these differences were no longer significant. Linear discriminant effect size analysis revealed a decrease in *Extibacter*,

deposited in the DNA database of the Japan sequence Read Archive (DRA) under the accession number DRA013989 (https://ddbj.nig.ac.jp/resource/sra-submission/DRA013989).

**Funding:** This work was supported in part by a grant from JSPS KAKENHI (19K09792), Tokyo, Japan (Japan Society for the Promotion of Science (jsps.go.jp)). TT was funded this grant. The funders had no role in study design, data collection and analysis, decision to publish, or preparation of the manuscript.

**Competing interests:** The authors have declared that no competing interests exist.

*Butyricicoccus*, *Megasphaera*, and *Parabacteroides* and an increase in *Anaerotaenia* in the PMDs group. The PSQ total score correlated with *Anaerotaenia*, *Extibacter*, and *Parabacteroides*. Multiple regression analysis showed that *Parabacteroides* and *Megasphaera* negatively predicted the PSQ total score.

## Conclusion

The properties of the gut microbiota are associated with premenstrual symptoms.

## Introduction

Premenstrual symptoms encompass a range of psycho-physical symptoms observed before menstruation, which interfere with the quality of life of many women between menarche and menopause [1–3]. In epidemiologic surveys, the prevalence of premenstrual symptoms is high (80%–90%) [4]. As a disease, it has been classified as premenstrual syndrome (PMS) by the field of obstetrics and gynecology and as premenstrual dysphoric disorder (PMDD) by the field of psychiatry, but recently both have been recognized by the inclusive term, premenstrual disorders (PMDs) [5]. Various causes have been suggested, including hormonal changes, serotonergic dysfunction, impaired gamma-aminobutyric acid (GABA) function, stress, and poor lifestyle habits such as longer durations of internet use and shorter sleep durations, but the precise pathophysiology of PMDs remains unknown [6–9].

As "another organ", the gut microbiota exhibits complex interactions with the immune, metabolic, and endocrine systems through the host's intestinal epithelium, and maintains a delicate balance [10–12]. Many important diseases such as obesity, metabolic disease, cardiovascular disease, inflammatory diseases, and brain disorders are found to be associated with differences in the gut microbiome [13, 14]. Molecular methods, especially amplicon sequence analysis using next generation sequencing, have enabled us to detect the diversity and composition of the gut microbiota in detail.

Recently, the association between the microbiota and brain function, such as in the case of major depressive disorder (MDD), has been extensively studied [15–18]. The gut microbiota communicates with the brain through neuroendocrine, neuroimmune, and neural pathways, and this is known as the gut microbiota–brain axis [14, 19–21]. According to clinical data, MDD is associated with low-grade inflammation (C reactive protein (CRP) >3 mg/L) [22, 23]. One possible mechanism may be microbiota-related inflammation, or so-called "leaky gut" [24]. A dysfunctional intestinal barrier may permit the translocation of Gram-negative bacteria and the bacterial endotoxin (lipopolysaccharide, LPS) from the gut microbiota to the bloodstream [24]. Bacterial translocation induces LPS-binding protein (LBP) and soluble CD14 (sCD14) that potentiates inflammation through Toll-like receptor (TLR)-4 and NF-κB activation [24]. LBP and sCD14 may be markers of endotoxemia and are reported to be associated with depressive symptoms [25, 26].

PMDs are linked to various mood and behavioral symptoms, which overlap with MDD. For treatment, serotonin reuptake inhibitors are recommended and commonly prescribed for both PMDs and MDD [2, 27–29]. Despite the commonality between PMDs and MDD, there have been no reports to date profiling the microbiota in PMDs. Our goal in this pilot study was to examine the possibility that characteristics of the gut microbiota may be related to the pathogenesis of PMDs. The aims of this study were to: (1) test the association between PMDs and bacterial translocation, (2) study the characteristics of the gut microbiota in women with

PMDs, and (3) study the association between the gut microbiota and premenstrual symptom severity.

## Materials and methods

### Ethics approval and informed consent

The study was carried out in accordance with the principles outlined in the Declaration of Helsinki. The trial protocol was approved by the Ethics Committee of Kindai University (approval number: 31–057). Written informed consent was obtained from all participants.

### Settings and participants

This study was conducted at an obstetrics and gynecology outpatient clinic in Osaka, Japan. Study participants were enrolled between September 2019 and August 2020. We recruited patients who wished to receive treatment for their premenstrual symptoms (P group, n = 27) and healthy volunteers who did not report serious premenstrual symptoms (N group, n = 29) (Fig 1). The inclusion criteria were: aged from 20 to 45 years, not receiving treatment for premenstrual symptoms, regular menstrual cycles (25 to 38 days), no neuropsychiatric disorders, and had not taken drospirenone-containing oral contraceptive (OC)s, antidepressants, herbal medicine, probiotics, or antibiotics for 4 weeks before study entry. Drospirenone-containing OCs were excluded because, unlike other conventional OCs, they have a proven therapeutic effect on PMDs [30]. In addition, the inclusion criteria for the P group included at least one or more of the symptoms listed in the premenstrual symptoms questionnaire (PSQ) showing a moderate or higher level, and for the N group included no symptoms listed in the PSQ showing a moderate or higher level. In total, the microbiotas of 56 patients were analyzed. None of these patients had a history of inflammatory bowel disease, irritable bowel disease, malignancy, any gastrointestinal tract surgery, or diabetes, and all were OC non-users. According to the criteria for PMDs defined by the International Society of Premenstrual Disorder, there is no regulation on the number of premenstrual symptoms, but marked interference to the patient's social life by the premenstrual symptoms is essential [5]. For an accurate diagnosis of PMD, it is recommended to keep a symptoms diary for two prospective periods, but this is not common in general gynecological practice in Japan. In this study, we further selected suspected cases of PMD (PMDs group) (n = 21) from the P group as those who experienced interference to their social life by the premenstrual symptoms. We also selected a control group (n = 22) from the N group who did not experience interference to their social life by premenstrual symptoms. The seven patients excluded from the N group had multiple mild premenstrual symptoms and mild interference to their social life due to these symptoms, as assessed by the PSQ.

### Questionnaire

**The premenstrual symptoms questionnaire.** In this study, we used the PSQ, developed in our previous study [31], to evaluate the severity of premenstrual symptoms. The PSQ has been found to have high reliability and validity [32].

The PSQ asks, "Within the last 3 months, have you experienced the following premenstrual symptoms starting during the week before menses and stopping a few days after the onset of menses?" The premenstrual symptoms listed are as follows: (i) depressed mood, (ii) anxiety or tension, (iii) tearfulness, (iv) anger or irritability, (v) decreased interest in work, home, or social activities, (vi) difficulty concentrating, (vii) fatigue or lack of energy, (viii) overeating or food cravings, (ix) insomnia or hypersomnia, (x) feeling overwhelmed, and (xi) physical

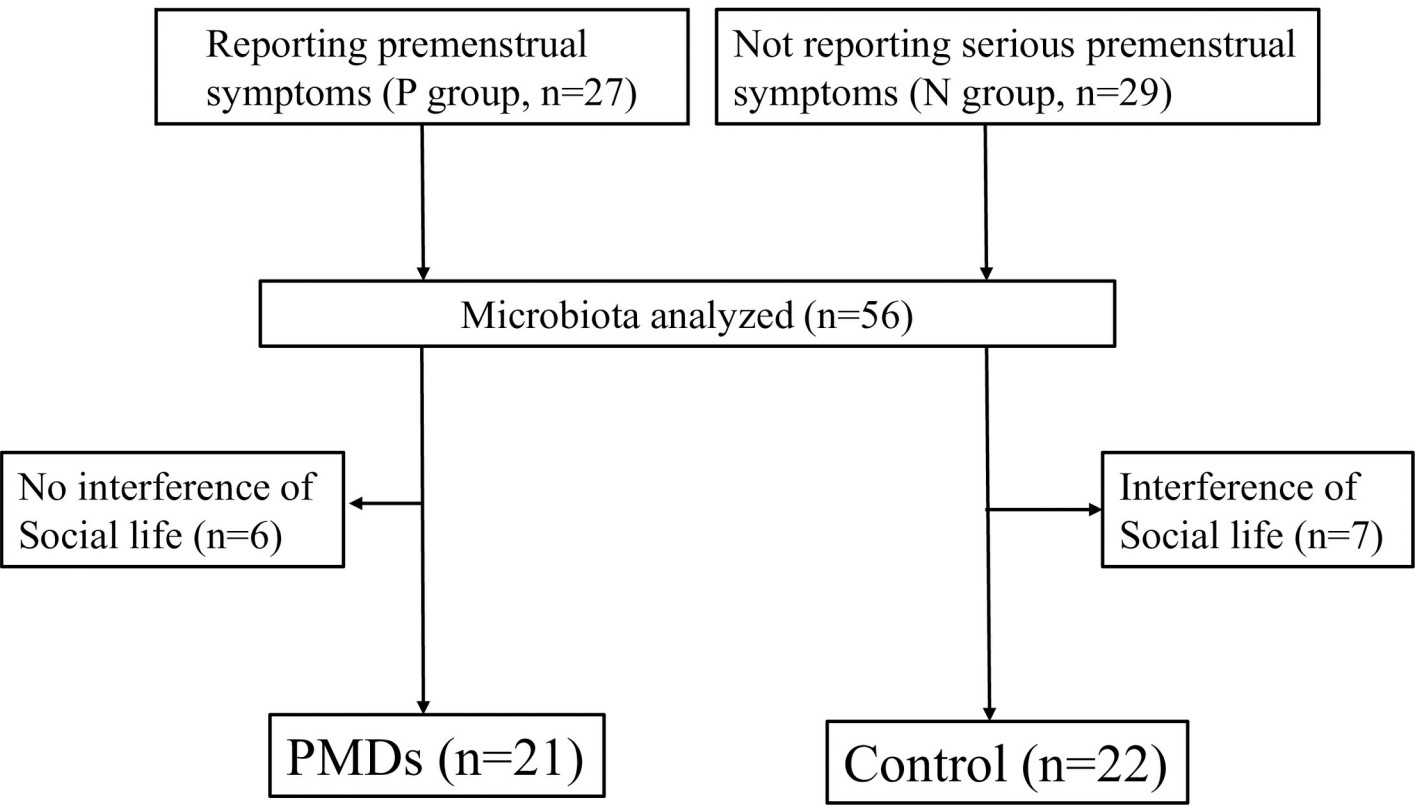

**Fig 1. Flow chart of the study design. Abbreviation:** PMDs, premenstrual disorders.

symptoms such as tender breasts, feeling of bloating, headache, joint or muscle pain, or weight gain. These 11 symptoms are listed in the diagnostic criteria for PMDD in the *Diagnostic and Statistical Manual of Mental Disorders* (DSM). Furthermore, the PSQ asks whether these symptoms interfere with (a) work efficiency or productivity, or home responsibilities; (b) social activities; or (c) relationships with coworkers or family. The participants were asked to rate the severity of premenstrual symptoms and the interference with daily activities resulting from these symptoms as *1—Not at all*, *2—Mild*, *3—Moderate*, or *4—Severe*. The total scores of the PSQ were calculated as the sum of the 14 items. The PSQ total score was applied for the evaluation of premenstrual symptoms severity and ranges from 14 to 56.

**Menstrual pain intensity.** A numerical rating scale (NRS) was applied for the evaluation of menstrual abdominal pain intensity. On the NRS, study participants rate their severity of menstrual pain from 0 (*no pain*) to 10 (*maximum pain you can imagine*).

**Evaluation of basic attributes.** For each participant, we also collected information about age, parity, body weight, height, age at menarche, total sleep duration, smoking (yes or no), drinking (yes or no), regular exercise (yes or no), diarrhea or constipation. Body mass index $(kg/m^2)$ was calculated by dividing weight in kilograms by height in meters squared.

## Measurement of inflammatory markers in blood samples

Blood samples were collected immediately after the PSQ and NRS assessment. Serum-separated samples were stored at −20˚C until further analysis. Frozen samples were transferred to Filgen, Inc. (Nagoya, Japan) and CRP, soluble CD14 (sCD14), and lipopolysaccharide binding protein (LBP) were analyzed using the Human Magnetic Luminex Assay (R&D Systems, Inc.,

Minneapolis, MN, USA) according to the manufacturer's protocols. All specimens were measured in triplicate and the average was obtained.

## Analysis of the microbiota

All subjects were instructed to collect a stool sample at home. The participants scraped the surface of their stool with a swab after defecation and collected stool specimens in sampling tubes containing guanidine thiocyanate solution (Techno Suruga Laboratory Co. Ltd., Shizuoka, Japan). The DNA preservation solution ensures that the DNA of the stool specimen in the sampling tube is stable for at least one month at room temperature or at 4˚C. The specimens were shipped by the subjects to our laboratory the same day using an express courier service, arriving no later than 48 hours after specimen collection. The collected specimens were sent to Techno Suruga Laboratory using a courier service at 4˚C. The samples were stored at 4˚C before DNA extraction. Extracted bacterial DNA was subjected to amplicon sequence analysis using the MiSeq system (Illumina, San Diego, CA, USA) by Techno Suruga Laboratory, as described previously [33]. Then, DNA was extracted using an automated DNA isolation system (GENE PREP STAR PI-480, Kurabo, Osaka, Japan), with 200 μL of distilled water being included as a negative control sample. The V3-V4 regions of Bacterial and Archaea 16S rRNA were amplified from the extracted DNA using the Pro341F/Pro805R primers and the dual-index method [33, 34], a negative control sample was also included, and the amplicons were visualized by electrophoresis. Barcoded amplicons were paired-end sequenced using a 2×284-bp cycle and the MiSeq system with MiSeq Reagent Kit version 3 (600 Cycle) chemistry. The quality of the paired-end sequencing reads was checked using the FASTX-Toolkit [35]. Paired-end sequencing reads were merged using the fastq-join program with default settings [36]. Only joined-reads that had a quality value score (QC) of $\geq 20$ for more than 99% of the sequence were extracted using the FASTX-Toolkit [35]. Chimeric sequences were deleted with usearch6.1 [37, 38]. Bacterial and Archaea species identification from sequences was performed using the Metagenome@KIN Ver 2.2.1 analysis software (World Fusion, Japan) and the TechnoSuruga Lab Microbial Identification database DB-BA 13.0 (TechnoSuruga Laboratory, Japan) with homology of $\geq 97\%$ [39]. The relative abundance of each bacterium at the phylum and genus level was calculated.

The 16S rRNA data were also analyzed with Quantitative Insights into Microbial Ecology (QIIME) 2.0 ver. 2020.6 [40]. Quality filtering and chimeric sequences were filtered using DADA2 (Divisive Amplicon Denoising Algorithm 2) denoise-single plugin ver. 2017.6.0 with default option [41]. Taxonomy was assigned using Greengenes database ver. 13.8 based on an average percent identity of 99% [42] by training a Naive Bayes classifier using the q2-feature-classifier plugin. To analyze beta diversity, weighted unifrac distance metrics were used. We used principal coordinates analysis (PCoA) to show the pattern of differences in the PMDs group and the control group. Alpha diversity was calculated by the Chao 1 [43], Shannon [44], and Simpson indices [45]. To further investigate the differences in abundance between the PMDs group and control group at the genus level, linear discriminant effect size analysis (LEfSe) was performed through the Huttenhower Lab Galaxy Server [46]. LEfSe was performed under the following conditions: the alfa value for the Kruskal–Wallis test was 0.05 and the threshold for the logarithmic linear discriminant analysis (LDA) score for a discriminative feature was 2.0.

## Statistical analysis

For continuous variables, normally distributed data were expressed as means and standard deviations and were analyzed by the Student's *t*-test, while non-normally distributed data were

expressed as medians and interquartile ranges and were analyzed by the Wilcoxon signed-rank test. For the Student's $t$-test, effect size was measured using $r$, and $r$ was calculated by the following formula ($r = \sqrt{\frac{t2}{t2+df}}$). For the Wilcoxon signed-rank test, effect size was measured using $r$, and $r$ was calculated by the following formula ($r = Z/\sqrt{N}$). For categorical variables, proportions were calculated and analyzed by Fisher's exact test. Effect size was measured using Cramer's V. The effect sizes of 0.10, 0.30, and 0.50 were judged as small, medium, and large, respectively [47].

For the relative abundance analysis of gut microbiota, multiple comparisons were adjusted using false discovery rate (FDR) correction. An FDR-adjusted $P$ value ($q$ value) < 0.20 was set as the cut-off [48].

Correlations between gut microbiota abundance and PSQ total score were analyzed using Spearman's rank correlation coefficient. Multiple regression analysis was used to explore the association between the microbiota and the PSQ total score. Variables that were predictive at a $P$ value less than 0.20 were introduced into the stepwise model.

Statistical analyses were performed using JMP Pro 15.1.0 (SAS, Cary, NC, USA), except for the relative abundance analysis of the gut microbiota, for which SAS 9.4 (SAS) was used. Statistical significance was set at $P$ < 0.05 (for two-tailed tests).

## Results

The characteristics of the study population are presented in Table 1.

The severity of menstrual pain was stronger and the PSQ total score was higher in the PMDs group than in the control group ($P$ < 0.0001).

The differences in endotoxin biomarkers between the two groups are presented in Table 2.

There was no significant difference between the expression levels of endotoxin biomarkers in the PMDs group and the control group.

Next, we analyzed alpha and beta diversity in the two groups (Fig 2). Regarding alpha diversity, there were no significant differences between the PMDs group and the control group as determined by the Chao 1, Shannon, and Simpson indices ($P$ = 0.430, 0.423, and 0.308,

**Table 1. Characteristics of the study participants.**

| Characteristic | Total (n = 56) | PMDs (n = 21) | Control (n = 22) | P |
|---|---|---|---|---|
| Age (years), median (IQR) | 27.5 (23.0–35.0) | 26.0 (23.0–31.0) | 26.0 (22.8–37.5) | 0.394[a] ($r$ = 0.130) |
| Parity, median (IQR) | 0 (0–1.8) | 0 (0–2.0) | 0 (0–0) | 0.073[a] ($r$ = 0.273) |
| Age at menarche (years), median (IQR) | 12.0 (11.0–13.0) | 12.0 (11.0–13.0) | 12.5 (11.0–14.0) | 0.397[a] ($r$ = 0.129) |
| BMI (kg/m$^2$), median (IQR) | 21.7 (19.3–23.1) | 21.2 (19.1–22.7) | 20.4 (19.0–23.4) | 0.799[a] ($r$ = 0.039) |
| Menstrual pain intensity, median (IQR) | 4.5 (3.0–7.0) | 8.0 (5.5–8.0) | 3.0 (1.0–4.0) | <0.0001[a] ($r$ = 0.656) |
| Total sleep duration (hours), median (IQR) | 6.0 (6.0–7.0) | 6.0 (6.0–7.0) | 6.3 (6.0–7.0) | 0.207[a] ($r$ = 0.192) |
| Smoker, n (%) | 9.0 (16.1) | 1.0 (4.8) | 6.0 (27.3) | 0.095[b] ($V$ = 0.305) |
| Drinker, n (%) | 19.0 (33.9) | 6.0 (28.6) | 9.0 (40.9) | 0.526[b] ($V$ = 0.129) |
| Regular exercise, n (%) | 7.0 (12.7) | 3.0 (15.0) | 1.0 (4.6) | 0.333[b] ($V$ = 0.178) |
| Diarrhea or constipation, n (%) | 23.0 (41.1) | 10.0 (47.6) | 7.0 (31.8) | 0.358[b] ($V$ = 0.162) |
| PSQ total score, median (IQR) | 22.0 (18.0–30.0) | 32.0 (28.5–40.5) | 17.0 (15.8–18.3) | <0.0001[a] ($r$ = 0.844) |

**Abbreviations:** PMDs, premenstrual disorders; IQR, interquartile range; BMI, body mass index; SD, standard deviation; PSQ, premenstrual symptoms questionnaire; V, Cramer's V

[a] Wilcoxon signed-rank test

[b] Fisher's exact test

**Table 2. Comparison of the endotoxin biomarkers present in the PMDs group and the control group.**

| Characteristic | PMDs (n = 21) | Control (n = 22) | P |
|---|---|---|---|
| CRP (ng/ml), median (IQR) | 308.2 (117.0–477.6) | 242.3 (61.1–980.6) | 0.743[a] (r = 0.050) |
| s-CD14 (ng/ml), mean (SD) | 918.2 (37.7) | 912.7 (36.8) | 0.917[b] (r = 0.016) |
| LBP (ng/ml), median (IQR) | 5859.6 (5213.7–6579.0) | 5404.2 (4506.6–7312.7) | 0.536[a] (r = 0.093) |

**Abbreviations:** PMDs, premenstrual disorders; CRP, C reactive protein; IQR, interquartile range; s-CD14, soluble cluster of differentiation 14; SD, standard deviation; LBP, lipopolysaccharide binding protein

[a] Wilcoxon signed-rank test

[b] Student's *t*-test

respectively). Regarding beta diversity, as analyzed by PCoA, a significant difference was detected between the PMDs group and the control group (R = 0.062, P = 0.027).

The relative abundance of organisms in the gut microbiota was compared between the PMDs group and the control group (Fig 3). At the phylum level, the PMDs group possessed fewer *Bacteroidetes* than the control group (P = 0.015, q = 0.136) (Fig 3A). We further analyzed the *Firmicutes/Bacteroidetes* (F/B) ratio, but this did not differ significantly between the two groups (P = 0.111). At the genus level, the PMDs group had a lower prevalence of *Butyricicoccus*, *Extibacter*, *Megasphaera*, *Parabacteroides*, and "Not determined" (P = 0.037, 0.018, 0.028, 0.039 and 0.033, respectively), and a higher prevalence of *Anaerotaenia* (P = 0.017) than the control group; however, after FDR correction, this significance was lost (Fig 3B).

Furthermore, we analyzed the characteristic gut microbiota differences in the PMDs group and the control group by LEfSe (Fig 4A and 4B). At the genus level, *Anaerotaenia* was enriched in the PMDs group, whereas *Extibacter*, *Butyricicoccus*, "Not determined", *Megasphaera*, and *Parabacteroides* were enriched in the control group (Fig 4B).

Next, we analyzed the association between the abundance of organisms in the gut microbiota and the severity of premenstrual symptoms (Table 3).

At the genus level, the PSQ total score was positively associated with *Anaerotaenia*, and negatively associated with *Extibacter* and *Parabacteroides*. Multiple regression analysis showed

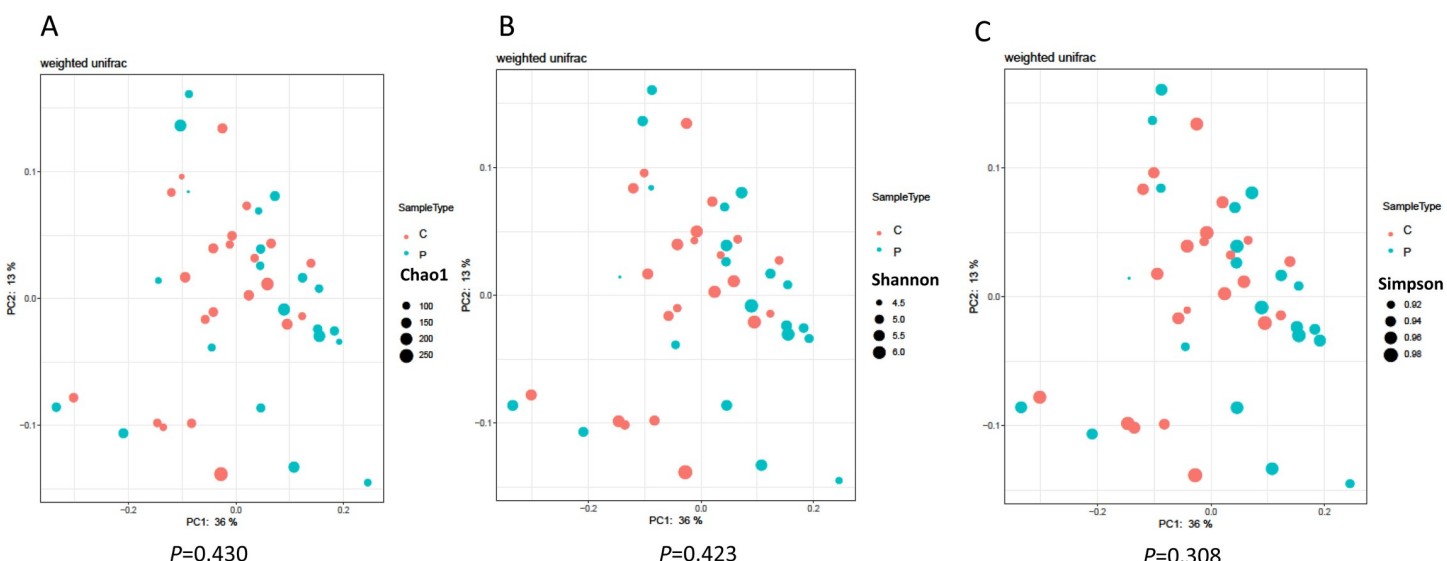

**Fig 2. Principal coordinates analysis plot comparing sample distribution between the PMDs group and the control group.** Each blot shows data for alpha diversity, the Chao 1 index (A), the Shannon index (B), and the Simpson index (C). **Abbreviations:** C, control group; P, premenstrual disorders group.

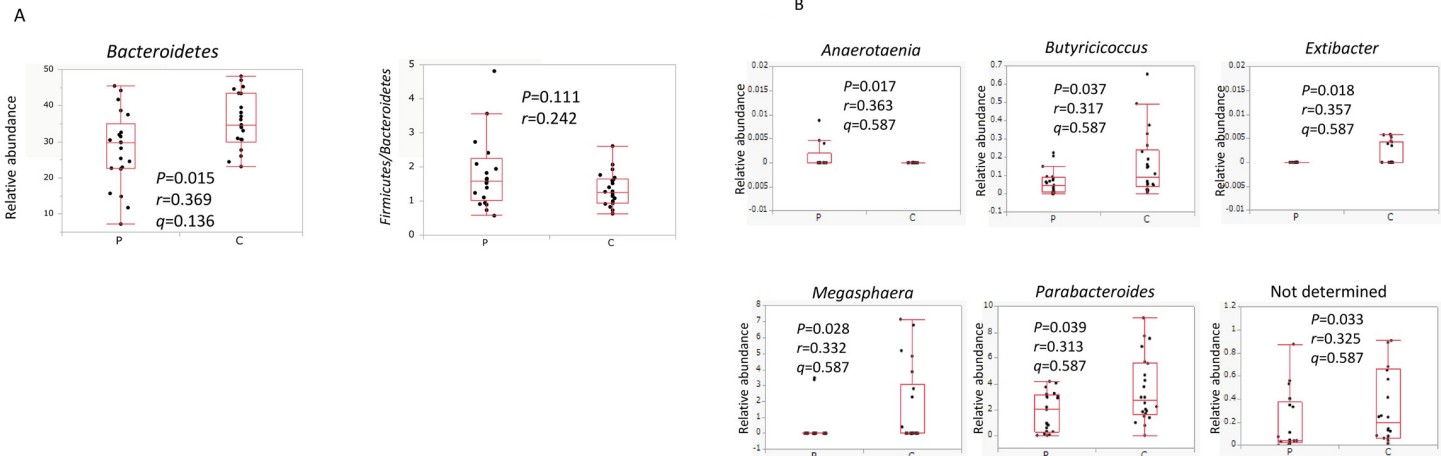

**Fig 3. Comparison of the gut microbiotas between the PMDs group and the control group.** The relative abundance of each taxon in the gut microbiota was compared. (A) At the phylum level, only Bacteroidetes was significantly less abundant in the PMDs group than in the control group. (B) The abundance of genus-level bacteria was significantly different between the PMDs group and the control group. The Wilcoxon signed-rank test was used to compare differences between the two groups ($P < 0.05$). **Abbreviations:** PMDs, premenstrual disorders; P, premenstrual disorders group; C, control group.

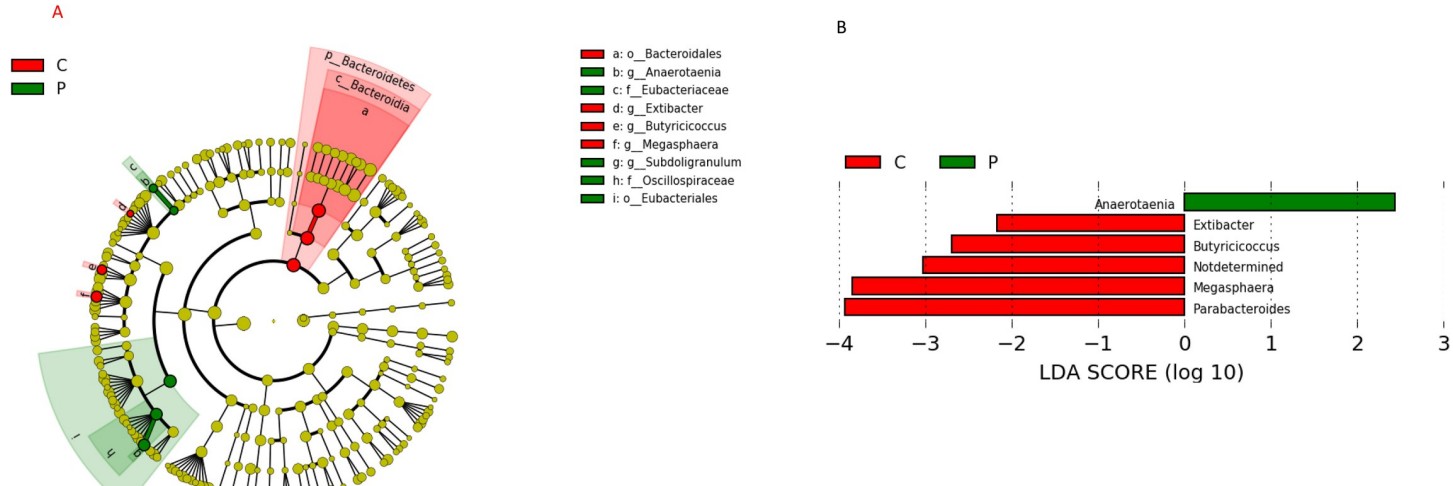

**Fig 4. Linear discriminant effect size analysis to distinguish the differential microbiota between the PMDs group and control group.** (A) Cladogram showing the most differentially abundant taxa between the PMDs group (P) and control group (C). Taxa enriched in the PMDs group are indicated in green and taxa enriched in the control group are indicated in red. The brightness of each dot is proportional to the respective effect size. (B) Comparison of the LDA scores between the P and C groups at the genus level. **Abbreviations:** LDA, logarithmic linear discriminant analysis; PMDs, premenstrual disorders; P, premenstrual disorders group; C, control group.

**Table 3. Correlation analysis between the gut microbiota abundance and the PSQ total score (n = 56).**

|  | **R** | **P** |
|---|---|---|
| *Anaerotaenia* | 0.292 | 0.029 |
| *Extibacter* | −0.410 | 0.002 |
| *Parabacteroides* | −0.342 | 0.010 |

**Abbreviations:** PSQ, premenstrual symptoms questionnaire

that the PSQ total score was negatively associated with *Parabacteroides* and *Megasphaera* (Table 4).

Variance inflation factor analysis showed that multicollinearity was not present for premenstrual symptoms in this model.

## Discussion

To our knowledge, this is the first report to investigate the association between the gut microbiota and premenstrual symptoms. By amplicon sequencing, we detected a difference in the gut microbiota between the PMDs group and the control group. Furthermore, we identified several organisms within the gut microbiota that were significantly associated with the severity of premenstrual symptoms.

In the case of MDD, increased bacterial translocation has been suggested to play a role in the inflammatory pathophysiology [24]. In this study, rather than directly assessing bacterial translocation by analyzing the presence of gut bacteria in the blood, we instead indirectly assessed bacterial translocation by measuring the levels of inflammatory factors CRP, LBP, and sCD14. LBP and sCD14 are produced in response to bacterial translocation and are proposed markers of endotoxemia [24]. Given the exploratory nature of this study, we chose the indirect method of evaluation as a simpler method for this initial investigation. According to our data, these indicators of inflammation did not show any correlation with PMDs. MDD and PMDs are closely related in terms of their clinical symptoms. However, the duration of symptoms is distinct between these two disorders, with MDD symptoms being permanent and PMD symptoms being temporary. This may explain the difference in the degree of inflammation between MDD and PMDs.

There was no significant difference in alpha diversity between the PMDs group and the control group. According to data from meta-analysis of MDD patients, there was no significant difference in alpha diversity between MDD patients and controls [18]. In other diseases, such as inflammatory bowel disease, obesity, and metabolic diseases, decreased microbiota diversity was suggested to be associated with the development of these diseases [49, 50]. This may explain some of the common pathology of MDD and PMDs.

At the phylum level, *Bacteroidetes* was less abundant in the PMDs group than in the control group. Low *Bacteroidetes* levels have been reported to be associated with obesity and MDD [51, 52]. Furthermore, obesity was reported to be a risk factor for PMDs and MDD [53, 54]. Considering that, in this study, no significant difference was found in BMI between the PMDs group and control group, obesity did not seem to be a confounding factor.

At the genus level, our data indicated that, in general, decreased levels of *Butyricicoccus*, *Extibacter*, *Megasphaera*, and *Parabacteroides* were associated with PMDs. These gut

**Table 4. Multiple regression analysis calculating the associations between the microbiota and the PSQ total score (n = 56).**

| | β | 95% CI | P | Standardized β | VIF |
|---|---|---|---|---|---|
| *Blautia* | 0.36 | −0.07 to 0.79 | 0.10 | 0.23 | 1.24 |
| *Faecalibacterium* | −0.33 | −0.82 to 0.17 | 0.19 | −0.17 | 1.07 |
| *Parabacteroides* | −1.31 | −2.43 to -0.18 | 0.02 | −0.30 | 1.14 |
| *Ruminococcus* | 1.15 | −0.07 to 2.36 | 0.06 | 0.25 | 1.27 |
| g_*Lachnospiraceae bacterium KNHs209_incertae_sedis* | −2.65 | −5.36 to 0.06 | 0.06 | −0.24 | 1.10 |
| *Megasphaera* | −1.56 | −2.89 to -0.24 | 0.02 | −0.31 | 1.19 |

$R^2$ = 0.29

**Abbreviations:** PSQ, premenstrual symptoms questionnaire; β, regression coefficient; CI, confidence interval; VIF, variance inflation factor

microorganisms were different from those reported to be decreased in MDD patients compared with non-MDD patients [18].

Among the gut microbiota, *Butyricicoccus* is a butyrate-producing beneficial bacterium and *Megasphaera* metabolizes lactate to butyrate [55]. In animal models, butyrate-treated mice possessed an increased amount of brain-derived neurotrophic factor (BDNF), which is essential for nerve cell growth and has been linked to antidepressant effects [56]. Decreased levels of butyrate-producing bacteria, such as *Butyricicoccus* and *Megasphaera*, may be involved in the pathology of PMDs.

Decreased levels of *Butyricicoccus* have been reported in postpartum depressive disorder (PPD) [57]. PPD is a unique subtype of MDD, for which the precise pathogenesis remains unknown. During pregnancy, women possess high levels of estrogen and progesterone because of the presence of the placenta and fetus. Dramatic hormonal fluctuations occur after delivery, which is thought to be one of the causes of PPD [58]. The hormonal fluctuations observed with PPD are the same as for PMDs, and premenstrual symptoms have been proposed as a risk factor for PPD [59]. Decreased levels of *Butyricicoccus* may explain some of the common pathology of PPD and PMDs.

*Parabacteroides* is reported to be beneficial in protecting against multiple sclerosis [60], seizures [61], metabolic dysfunctions [62], and colon tumors [63]. Regarding the gut microbiota–brain axis, *Parabacteroides* was reported to be a GABA-producer according to the results of GABA-dependent co-culture assays and *in silico* analyses [64]. In a seizure mouse model, the anti-seizure effects of a ketogenic diet were analyzed in the gut microbiota [61]. In this study, *Parabacteroides* was shown to modulate brain GABA levels. Because impaired GABA function is one of the possible causes of PMDs [7], decreased levels of *Parabacteroides* may be involved in PMD pathology.

Our study had several limitations. The main limitation was that the study was cross-sectional in design. It was therefore impossible to determine causality between premenstrual symptoms and the gut microbiota changes. To clarify the causality, further longitudinal studies are required. Second, our study had a small sample size; however, we believe this was appropriate for a pilot study. Third, we selected the participants from outpatients seeking care. This is a highly select population, which could have led to data bias. Fourth, we selected the PMDs group without prospective daily charting over two consecutive symptomatic cycles, which is recommended by the DSM [30]. However, prospective daily charting is difficult in clinical settings. According to a report from the USA, only 11.5% of physicians reported routinely monitoring two consecutive symptomatic cycles [65]. Fifth, the timing of blood and stool sample collection was performed without considering the menstrual cycle. There is a possibility that the gut microbiota may fluctuate depending on the stage of the menstrual cycle, and sampling at a consistent stage of the menstrual cycle may be necessary in future investigations. Sixth, the study was conducted only in Japan, which might limit the generalization of the findings to the other countries, especially western countries. Finally, the significant difference in the relative abundance of the gut microbiota between the PMDs group and the control group was lost after FDR correction. Considering that the results of LEfSe showed the same pattern of differences in abundance between the PMDs group and the control group, these differences in the gut microbiota between the two groups are likely to be meaningful. Further studies considering these factors are needed to confirm the reliability of these findings. Consecutive prospective evaluations using the Daily Record of Severity of Problem (DRSP) are recommended for accurate assessment of premenstrual symptoms [30], and it is possible to use the Japanese version of the DRSP, for which the validity and reliability have been confirmed [66]. Using DRSP assessment in a large-scale prospective study of the general public, and performing blood and stool collection at specific times during the follicular and luteal phases, respectively, would be expected to provide more definitive results.

Despite these limitations, differences in the gut microbiota between PMD patients and healthy individuals may be applied as biomarkers for diagnosis in the future.

## Conclusions

The present study showed that gut microbial properties were associated with premenstrual symptoms. Decreased levels of *Parabacteroides* and *Megasphaera* are a characteristic feature of PMD patients and are negatively associated with the severity of premenstrual symptoms.

## Supporting information

**S1 File. Data for Table 1.**
(XLSX)

**S2 File. Data for Table 2.**
(XLSX)

**S3 File. Phylum-level operational taxonomic units of the fecal microbiota.**
(XLSX)

**S4 File. Genus-level operational taxonomic units of the fecal microbiota.**
(XLSX)

## Acknowledgments

We thank Edanz (https://jp.edanz.com/ac) for editing a draft of this manuscript.

## Author Contributions

**Conceptualization:** Takashi Takeda.

**Data curation:** Takashi Takeda, Kana Yoshimi, Sayaka Kai, Keizo Hiramatsu.

**Formal analysis:** Takashi Takeda, Keiko Yamada.

**Funding acquisition:** Takashi Takeda.

**Investigation:** Takashi Takeda, Sayaka Kai, Genki Ozawa.

**Methodology:** Takashi Takeda, Genki Ozawa.

**Project administration:** Takashi Takeda.

**Resources:** Takashi Takeda, Genki Ozawa, Keizo Hiramatsu.

**Software:** Takashi Takeda, Genki Ozawa, Keiko Yamada.

**Supervision:** Takashi Takeda, Kana Yoshimi, Keiko Yamada, Keizo Hiramatsu.

**Validation:** Takashi Takeda.

**Visualization:** Takashi Takeda, Genki Ozawa.

**Writing – original draft:** Takashi Takeda, Kana Yoshimi, Sayaka Kai, Genki Ozawa, Keizo Hiramatsu.

**Writing – review & editing:** Takashi Takeda, Kana Yoshimi, Sayaka Kai, Genki Ozawa, Keiko Yamada, Keizo Hiramatsu.

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
