## [Decision Letter · Decision Letter 0]

2 Nov 2021

PONE-D-21-30216Characteristic of the gut microbiota in women with premenstrual symptoms : a cross-sectional studyPLOS ONE

Dear Dr. Takeda,

Thank you for submitting your manuscript to PLOS ONE. After careful consideration, we feel that it has merit but does not fully meet PLOS ONE’s publication criteria as it currently stands. Therefore, we invite you to submit a revised version of the manuscript that addresses the points raised during the review process.

The reviewers identified multiple issues related to study design, methodology, and interpretation that the authors need to address. I would like to highlight the following ones which are particularly critical for the authors to address in the revision for the manuscript to be suitable for publication:

1) There are extensive problems with the clarity of the writing (many specific examples are provided by the reviewers). I recommend the use of an English language writing service to assist with revisions.

2) More precise description of the study population is required (e.g. distinction between PMS vs. PMDD, clarification of "social disturbance" as a factor for selecting the two study groups, which subjects were included in each of the microbiome analyses that are shown)

3) Statistical analyses need to be corrected for multiple hypothesis testing (and if significance is lost after FDR correction, this needs to be addressed as a limitation)

4) Deidentified data needs to be made available per PLOS ONE policy

 rebuttal letter that responds to each point raised by the academic editor and reviewer(s). You should upload this letter as a separate file labeled 'Response to Reviewers'.A marked-up copy of your manuscript that highlights changes made to the original version. You should upload this as a separate file labeled 'Revised Manuscript with Track Changes'.An unmarked version of your revised paper without tracked changes. You should upload this as a separate file labeled 'Manuscript'.

We look forward to receiving your revised manuscript.

Kind regards,

Jonathan Jacobs

Academic Editor

PLOS ONE

Journal Requirements:

4.  We noticed you have some minor occurrence of overlapping text with the following previous publication, which needs to be addressed:

- https://www.dovepress.com/psychometric-testing-of-the-premenstrual-symptoms-questionnaire-and-th-peer-reviewed-fulltext-article-IJWH

The text that needs to be addressed involves the first paragraph of the Introduction. 

In your revision ensure you cite all your sources (including your own works), and quote or rephrase any duplicated text outside the methods section. Further consideration is dependent on these concerns being addressed.

Reviewers' comments:

Reviewer's Responses to Questions

**Comments to the Author**

1. Is the manuscript technically sound, and do the data support the conclusions?

Reviewer #1: Partly

Reviewer #2: No

2. Has the statistical analysis been performed appropriately and rigorously? 

Reviewer #1: Yes

Reviewer #2: No

3. Have the authors made all data underlying the findings in their manuscript fully available?

Reviewer #1: No

Reviewer #2: No

4. Is the manuscript presented in an intelligible fashion and written in standard English?

Reviewer #1: No

Reviewer #2: Yes

5. Review Comments to the Author

Reviewer #1: Takeda et al studied fecal microbiome using 16s rRNA sequencing in premenstrual disorder (PMD) patients compared to control women. They found differences in the composition of bacteria in women with PMD compared to control women. This is an interesting study and has attempted to address an important question. However, there are several issues pointed below, that need to be addressed.

1. The sample sizes in abstract and the main text are different (27 PMS and 29 controls in abstract vs 30 each in the main text). Additionally, it is unclear if the entire group was studied or the PMD subgroup was studied for microbiome-related analyses. The authors suggest that the subset of control subjects studied (N=22) did not have PMS or social disturbance. It is not clear what the authors mean by “social disturbance”. If it is one of the features of PMS, why were 8 controls (30-22) subjects included with “social disturbance”? Is there a microbial analysis performed in the larger group as well as in the smaller group? If not, the actual sample size of the study would be 22 and 21. It would help if the terms PMS, PMDD and PMD be described initially in the manuscript and only those abbreviations be used when referring the dataset consistently. For example, line 90, the recruitment was for “patients who wished to receive treatment”. If these were PMS, then it should be stated something on these lines - “Thirty patients with PMS were recruited …” and state how many had PMD or PMDD.

2. The entire Methods section of the abstract needs to be reworded for better clarity and correctness (for example, “complaining premenstrual symptoms”, which can be reworded as “experiencing ..”).

3. Aims stated on line 83 do not sufficiently capture the analysis and should be clearly stated.

4. If a correlation was tested for all genera, multiple testing corrections need to be applied and significance threshold needs to be implemented at an appropriate FDR cutoff (typically 5 or 10%) instead of a p-value cutoff.

5. Language and grammar need to be improved and space between words need to be fixed at several places in the abstract and the main manuscript. Some places are listed below

a. Line 139: should be “triplicate”.

b. Statistical analysis: “abnormally distributed” should be restated as “non-normally”; “Medium” should be “median”.

c. Line 248: Needs to be reworded.

d. Line 280: “High gonadal condition” is unclear.

6. 252: How did the authors measure “bacterial translocation”? The entire paragraph describing inflammation and bacterial translocation is unclear.

7. Asterisk usually refers to statistical significance. Please use another symbol in Table 1 to describe the tests used.

8. As per Figure 3B, control group had very low abundance of Anaerotaenia and Extibacter and PMD group had very low abundance of Megasphaera. It would be better to include points/jitter. Looks like it may be driven by outliers.

Reviewer #2: This paper presents data from a cross-sectional study on relationship between gut microbiota and premenstrual symptoms. The purpose of the study is innovative and interesting; however, multiple design and methodological flaws dampen my enthusiasm for the study. My detailed comments and suggestions are below.

Abstract:

1. The scope of the study can be clearer. Did the authors focus broadly on premenstrual symptoms? Premenstrual syndrome? or PMDD?

2. The term “social disturbance” is confusing. It sounds like social unrest. I believe the author mean disruption or interference of social life.

3. What type of microbial diversity did you refer to in the abstract?

4. It would be helpful to report P values and effect sizes when possible.

Data availability:

5. It is unclear why deidentified data underlying the findings cannot be made fully available. The openly sharing of deidentified data is important to safeguard the reproducibility of the study findings.

Background:

6. Again, the focus of the study needs to be clearer. Premenstrual syndrome? PMDD? Both PMS and PMDD? Premenstrual symptoms in general?

7. Line 60: Certain lifestyle factors have been found to be associated with premenstrual symptoms. However, it is imprecise to say that poor lifestyle habits cause the symptoms.

8. Line 4: What are the characteristics of the “systematic inflammation disorder”?

9. The rationale for selected blood biomarkers can be clearer. There are many measures for intestinal permeability and bacteria translocation. Have markers selected by the authors been linked to premenstrual symptoms in previous research?

10. Pre-specified hypotheses need to be clearly stated.

Methods:

11. Many possible comorbidities that are known to affect gut microbiome were neither considered in the design nor the the analysis (inflammatory bowel diseases, irritable bowel syndrome, diabetes). This is a major limitation of the study. Please acknowledge this in the discussion.

12. Similarly, is there information available regarding psycholocial comorbidities among participants?

13. The sample were selected patients seeking care at the outpatient ob/gyn clinic. This can be a highly selected population. Please acknowledge this as a limitation.

14. The antibiotic usage before sample collection was not clear.

15. What was the rationale for sample size?

16. Case and control need to be better defined. What is the rationale for case and control selection?

17. The case definition seems quite inconsistent with the DSM-3 criteria. The DSM criteria require to have at least 5 symptoms for PMDD diagnosis. In the study, the presence of one moderate or severe symptom will qualify the participants as cases. Also, the symptoms within the last 3 months can be different from most menstrual cycles (part of DSM-definition). These are major limitations of the study.

18. For menstrual pain intensity, which type of menstrual pain was assessed? Abdominal? Menstrual headache? Both?

19. When were stool and blood samples collected? Was menstrual cycle stage controlled? Sex hormones can potentially affect gut microbiome profile.

20. It is unclear if rectal swab or stool was collected. What was the length of time from sample collection to sample receipt? How was the sample stored?

21. For microbiome assays, were positive and negative controls used? If so, please describe. Also, what quality control measures were used?

22. For data analysis: What correction methods were used for multiple comparisons? If no correction method was used, please acknowledge this as a limitation.

Results:

23. For effect size measures from the LEfSE analysis, it would be helpful to show the plots in the results.

Discussion:

24. Please discuss and potential confounders (e.g., comorbid gastrointestinal and psychological factors) that may influence/confound the study results.

25. There are many different measures of bacterial translocation in the literature. Please comment the quality of the bacterial translocation measure used in the study. This will help readers interpret the findings related to blood biomarker measures.

26. Please acknowledge several limitations noted in my comments of the methods section.

27. The implications for future research can be clearer. What further research is needed to further investigate the mechanisms of gut microbiome- premenstrual symptom association?

6. PLOS authors have the option to publish the peer review history of their article (what does this mean?). If published, this will include your full peer review and any attached files.

Reviewer #1: **Yes: **Swapna Mahurkar-Joshi

Reviewer #2: No

---

## [Author Response · Author response to Decision Letter 0]

16 Dec 2021

Professor Emily Chenette

Editor-in-Chief 

PLOS ONE

 December 16, 2021

Dear Professor Chenette,

Thank you for reconsidering our manuscript entitled “Characteristics of the gut microbiota in women with premenstrual symptoms: A cross-sectional study” as an Original Research article for publication in PLOS ONE. We have revised our manuscript in response to the many helpful comments of the reviewers, as shown below. 

Academic Editor’s comments

 1) There are extensive problems with the clarity of the writing (many specific examples are provided by the reviewers). I recommend the use of an English language writing service to assist with revisions.

Response (Re): Our revised manuscript has been reviewed by a native English-speaking expert editor.

2) More precise description of the study population is required (e.g. distinction between PMS vs. PMDD, clarification of "social disturbance" as a factor for selecting the two study groups, which subjects were included in each of the microbiome analyses that are shown)

Re: We have added an explanation about the difference between PMS, PMDD, and PMDs on page 4, lines 57–60. We further added the criteria for PMDs and explained the importance of “social disturbance” (or “interference with social life”) in these criteria, as described on page 6, lines 115–118.

3) Statistical analyses need to be corrected for multiple hypothesis testing (and if significance is lost after FDR correction, this needs to be addressed as a limitation) 

Re: We applied FDR correction and added the data on the q values to Figure 3. We also added an explanation of this to the Materials and Methods section on page 11, lines 218–220, and added a new reference (47). In the original version, the JMP software was used for statistical analysis, but since FDR correction was not possible, the analysis was redone using the SAS software. As a result of this revision, the P value for the Wilcoxon signed-rank test changed slightly, so the results were revised in Figure 3 and on page 14, lines 274–275. In conducting this additional analysis, Keiko Yamada was added as a co-author. The difference in relative abundance of the gut microbiota between the PMDs group and the control group was lost after FDR correction and we have added an explanation of this on page 3, lines 40–41 and page 14, lines 275–276. Furthermore, we have added this to the limitations section on page 21, lines 391–396.

4) Deidentified data needs to be made available per PLOS ONE policy

Re: The data set supporting the results cannot be made accessible to the public because it contains potentially sensitive patient information. Based on the ethical guidelines in Japan, the ethics committee of Kindai University has imposed restrictions on the dissemination of the data in this study. The use of this data set requires approval by the ethics committee of Kindai University. In the case of a data access request, please contact the corresponding author: Takashi Takeda (take@med.kindai.ac.jp).

Journal Requirements:

Re: We have followed the PLOS ONE style as indicated.

2. In your Data Availability statement, you have not specified where the minimal data set underlying the results described in your manuscript can be found. PLOS defines a study's minimal data set as the underlying data used to reach the conclusions drawn in the manuscript and any additional data required to replicate the reported study findings in their entirety. All PLOS journals require that the minimal data set be made fully available. Upon re-submitting your revised manuscript, please upload your study’s minimal underlying data set as either Supporting Information files or to a stable, public repository and include the relevant URLs, DOIs, or accession numbers within your revised cover letter. For a list of acceptable repositories, please see http://journals.plos.org/ plosone/s/data-availability#loc-recommended-repositories. Any potentially identifying patient information must be fully anonymized.

Re: The data set supporting the results cannot be made accessible to the public because it contains potentially sensitive patient information. Based on the ethical guidelines in Japan, the ethics committee of Kindai University has imposed restrictions on the dissemination of the data in this study. The use of this data set requires approval by the ethics committee of Kindai University. In the case of a data access request, please contact the corresponding author: Takashi Takeda (take@med.kindai.ac.jp).

Re: We have moved the ethics statement to the Materials and Methods section as indicated.

4. We noticed you have some minor occurrence of overlapping text with the following previous publication, which needs to be addressed:

The text that needs to be addressed involves the first paragraph of the Introduction.

In your revision ensure you cite all your sources (including your own works), and quote or rephrase any duplicated text outside the methods section. Further consideration is dependent on these concerns being addressed.

Re: According to this comment, we have removed the overlapping text and correctly cited all sources on page 4, lines 54–64. 

Reviewer: 1

1. The sample sizes in abstract and the main text are different (27 PMS and 29 controls in abstract vs 30 each in the main text). Additionally, it is unclear if the entire group was studied or the PMD subgroup was studied for microbiome- related analyses. The authors suggest that the subset of control subjects studied (N=22) did not have PMS or social disturbance. It is not clear what the authors mean by “social disturbance”. If it is one of the features of PMS, why were 8 controls (30-22) subjects included with “social disturbance”? Is there a microbial analysis performed in the larger group as well as in the smaller group? If not, the actual sample size of the study would be 22 and 21. It would help if the terms PMS, PMDD and PMD be described initially in the manuscript and only those abbreviations be used when referring the dataset consistently. For example, line 90, the recruitment was for “patients who wished to receive treatment”. If these were PMS, then it should be stated something on these lines - “Thirty patients with PMS were recruited …” and state how many had PMD or PMDD.

Re: As pointed out by the reviewers, the definition of subjects was unclear in the original version. In the present study, there was no diagnosis of PMS, PMDD, or PMD in the strict sense of the word, only patients who came to the clinic for treatment of premenstrual symptoms. For an accurate diagnosis of PMS, PMDD, or PMD, it is necessary to keep a symptom diary for two prospective periods, but this is not common in general gynecological practice in Japan. We have added an explanation of this on page 6, lines 118–120. Groups P and N in the text were those who meet the inclusion and exclusion criteria, and the description of 30 women in each group was incorrect and has been revised to 27 women in P group and 29 women in N group as described on page 6, lines 104–106. In line with this revision, the description in Figure 1 has also been changed. According to the criteria for PMDs by the International Society of Premenstrual Disorder, there is no regulation on the number of premenstrual symptoms, but marked interference with the patient’s social life by the premenstrual symptoms is essential. According to this background, we selected suspected cases of PMD (the PMDs group) from the P group based on interference with their social life. We have added this explanation on page 6, lines 115–121, and page 7, line 122–124. We also corrected the expression “PMS” to “PMDs” on page 3, line 43. 

2. The entire Methods section of the abstract needs to be reworded for better clarity and correctness (for example, “complaining premenstrual symptoms”, which can be reworded as “experiencing ..”).

Re: According to this comment, we reworded “complaining premenstrual symptoms” to “experiencing premenstrual symptoms”, and “social disturbance” to “interference to their social life” on page 2, lines 27–29.

5. Language and grammar need to be improved and space between words need to be fixed at several places in the abstract and the main manuscript. Some places are listed below

a. Line 139: should be “triplicate”.

Re: According to this comment, “triple cate” has been corrected to “triplicate”. Our revised manuscript has also been reviewed by a native English-speaking expert editor.

b. Statistical analysis: “abnormally distributed” should be restated as “non-normally”; “Medium” should be “median”.

Re: According to this comment, we reworded “abnormally distributed” to “non-normally”, and “Medium” to “median”. 

c. Line 248: Needs to be reworded.

Re: According to this comment, we added “amplicon sequence analysis” to the explanation of next generation sequencing. In connection with this change, the description of next generation sequencing was revised on page 4, line 70, page 9, line 174 and page 17, line 323. 

d. Line 280: “High gonadal condition” is unclear.

Re: According to this comment, we reworded “high gonadal condition” to “high levels of estrogen and progesterone” on page 19, lines 362–363. 

6. 252: How did the authors measure “bacterial translocation”? The entire paragraph describing inflammation and bacterial translocation is unclear.

Re: Rather than directly assessing bacterial translocation by analyzing the presence of gut bacteria in the blood, we instead indirectly assessed bacterial translocation by measuring the levels of inflammatory factors CRP, LBP, and sCD14. LBP and sCD14 are produced in response to bacterial translocation and are proposed markers of endotoxemia. We added this explanation on page 17, lines 328–333. 

7. Asterisk usually refers to statistical significance. Please use another symbol in Table 1 to describe the tests used.

Re: We changed the asterisk to alphabetic characters in Table 1 and Table 2.

8. As per Figure 3B, control group had very low abundance of Anaerotaenia and Extibacter and PMD group had very low abundance of Megasphaera. It would be better to include points/jitter. Looks like it may be driven by outliers.

Re: We changed the graph display method to include points/jitter in Figure 3.

Reviewer: 2

Abstract:

1. The scope of the study can be clearer. Did the authors focus broadly on premenstrual symptoms? Premenstrual syndrome? or PMDD?

Re: The scope of the study was PMDs and premenstrual symptoms. We have clarified the text in the Abstract accordingly on page 2, lines 21–22.

2. The term “social disturbance” is confusing. It sounds like social unrest. I believe the author mean disruption or interference of social life.

Re: According to this comment, we have changed “social disturbance” to “interference to their social life” on page 2, lines 28 and 29 and page 7, line 122–124.

3. What type of microbial diversity did you refer to in the abstract?

Re: In the Abstract, we described alpha diversity and beta diversity. We have added this detail on page 2, lines 36.

4. It would be helpful to report P values and effect sizes when possible.

Re: We analyzed the effect sizes and have added these data to Tables 1 and 2. We also added an explanation of this analysis on page 10, lines 212–213 and page 11, lines 214–217. 

Data availability:

5. It is unclear why deidentified data underlying the findings cannot be made fully available. The openly sharing of deidentified data is important to safeguard the reproducibility of the study findings.

Re: The data set supporting the results cannot be made accessible to the public because it contains potentially sensitive patient information. Based on the ethical guidelines in Japan, the ethics committee of Kindai University has imposed restrictions on the dissemination of the data in this study. The use of this data set requires approval by the ethics committee of Kindai University. In the case of a data access request, please contact the corresponding author: Takashi Takeda (take@med.kindai.ac.jp).

Background:

6. Again, the focus of the study needs to be clearer. Premenstrual syndrome? PMDD? Both PMS and PMDD? Premenstrual symptoms in general?

Re: The scope of the study was PMDs and premenstrual symptoms. We have clarified this point on page 5, lines 89–92. 

7. Line 60: Certain lifestyle factors have been found to be associated with premenstrual symptoms. However, it is imprecise to say that poor lifestyle habits cause the symptoms.

Re: We agree and have modified the text accordingly on page 4, line 62–63. 

8. Line 4: What are the characteristics of the “systematic inflammation disorder”?

Re: This expression was inaccurate and has been corrected to “low-grade inflammation” on page 5, line 76. 

9. The rationale for selected blood biomarkers can be clearer. There are many measures for intestinal permeability and bacteria translocation. Have markers selected by the authors been linked to premenstrual symptoms in previous research?

Re: Although LBP and sCD14 have not been linked to premenstrual symptoms previously, they have been studied in relation to depressive symptoms. We have added the relevant references for this (25, 26) and an explanation on page 5, lines 82–83. 

10. Pre-specified hypotheses need to be clearly stated.

Re: We have added the hypotheses and aims of this study at the end of the Introduction on page 5, lines 87–92.

Methods:

11. Many possible comorbidities that are known to affect gut microbiome were neither considered in the design nor the the analysis (inflammatory bowel diseases, irritable bowel syndrome, diabetes). This is a major limitation of the study. Please acknowledge this in the discussion.

Re: These diseases were not considered in this study, and patients with these diseases were excluded. We have acknowledged this on page 6, lines 113–115.

12. Similarly, is there information available regarding psycholocial comorbidities among participants?

Re: We excluded any patients with neuropsychiatric disorders from the study. We have added this information to the inclusion criteria on page 6, line 108.

13. The sample were selected patients seeking care at the outpatient ob/gyn clinic. This can be a highly selected population. Please acknowledge this as a limitation.

Re: We have added this as a limitation on page 20, lines 380–382.

14. The antibiotic usage before sample collection was not clear.

Re: We specify the absence of antibiotic usage in the inclusion criteria on page 6, line 109. 

15. What was the rationale for sample size?

Re: Since there are no reports on the relationship between premenstrual symptoms and gut microbiota, we conducted a pilot study. Normally, we consider 20 to 30 cases to be appropriate for such an exploratory study; however, we have mentioned this as a limitation of the study on page 20, lines 379–380. 

16. Case and control need to be better defined. What is the rationale for case and control selection?

Re: There were two parts to our study. The first was to compare suspected PMD cases with control cases, and the second was to examine the relationship between the severity of premenstrual symptoms and the gut microbiota. This is clearly stated as the aim of the study in the last part of the Introduction on page 5, lines 89–92. In addition, the diagnosis of PMDs has been added to the Materials and Methods section on page 6, lines 115–120. 

17. The case definition seems quite inconsistent with the DSM-3 criteria. The DSM criteria require to have at least 5 symptoms for PMDD diagnosis. In the study, the presence of one moderate or severe symptom will qualify the participants as cases. Also, the symptoms within the last 3 months can be different from most menstrual cycles (part of DSM-definition). These are major limitations of the study.

Re: The current study targets PMDs and not PMS or PMDD. The differences between these disorders have been clarified in the Introduction on page 4, lines 57–60. In addition, the criteria for the diagnosis of PMD have been added to the Materials and Methods section on page 6, lines 115–120.

18. For menstrual pain intensity, which type of menstrual pain was assessed? Abdominal? Menstrual headache? Both?

Re: Menstrual pain usually refers to abdominal pain. We have clarified this on page 8, line 151.

 19. When were stool and blood samples collected? Was menstrual cycle stage controlled? Sex hormones can potentially affect gut microbiome profile.

Re: The timings of blood and stool collection were conducted without considering the menstrual cycle. We have added this as a limitation of the study on page 20, lines 386–389. 

20. It is unclear if rectal swab or stool was collected. What was the length of time from sample collection to sample receipt? How was the sample stored?

Re: Stool samples were collected by swabbing immediately after defecation. We have added this information on page 9, lines 170–171, and line 173.

21. For microbiome assays, were positive and negative controls used? If so, please describe. Also, what quality control measures were used?

Re: Negative controls were included as described on page 9, lines 177–178, and lines 180. We also describe the quality control measures on page 9, lines 183–184. 

22. For data analysis: What correction methods were used for multiple comparisons? If no correction method was used, please acknowledge this as a limitation.

Re: We applied FDR correction and have added the q-value data to Figure 3, along with an explanation in the Materials and Methods section on page 11, lines 218–220, and a new reference (47). In the original version, the JMP software was used for statistical analysis, but since FDR correction was not possible, the analysis was redone using the SAS software. As a result of this revision, the P value resulting from the Wilcoxon signed-rank test changed slightly, so the results were revised in Figure 3 and on page 14, lines 274–275. To conduct this additional analysis, Keiko Yamada was added as a co-author. The difference in the relative abundance of the gut microbiota between the PMDs group and the control group was lost after FDR correction and we have added this information on page 3, lines 40–41 and page 14, lines 275–276. We have also added this as a limitation on page 21, lines 391–396. 

23. For effect size measures from the LEfSE analysis, it would be helpful to show the plots in the results.

Re: We performed LEfSE through the Huttenhower Lab Galaxy Server, but we were unable to use the plots for the results from this server. 

Discussion:

24.Please discuss and potential confounders (e.g., comorbid gastrointestinal and psychological factors) that may influence/confound the study results.

Re: As stated in our response to comments 11 and 12, these potential confounders were not considered in our study. Furthermore, the timings of blood and stool collection did not take the menstrual cycle into consideration and this has been added as a limitation on page 20, lines 386–389. 

25.There are many different measures of bacterial translocation in the literature. Please comment the quality of the bacterial translocation measure used in the study. This will help readers interpret the findings related to blood biomarker measures.

Re: We have added an explanation of the blood biomarkers used on page 17, lines 328–333. 

26.Please acknowledge several limitations noted in my comments of the methods section.

Re: We have added a comprehensive list of the limitations of our study on page 20, lines 380–382, page 20, lines 386–389 and page 21, lines 391–396. 

27.The implications for future research can be clearer. What further research is needed to further investigate the mechanisms of gut microbiome- premenstrual symptom association?

Re: We have added the implications in terms of future research to the Discussion section on page 21, lines 397–403.

---

## [Decision Letter · Decision Letter 1]

10 Mar 2022

PONE-D-21-30216R1Characteristic of the gut microbiota in women with premenstrual symptoms : a cross-sectional studyPLOS ONE

Dear Dr. Takeda,

Thank you for submitting your manuscript to PLOS ONE. After careful consideration, we feel that it has merit but does not fully meet PLOS ONE’s publication criteria as it currently stands. Therefore, we invite you to submit a revised version of the manuscript that addresses the points raised during the review process.

The primary remaining issue is data availability. Please see the policy of PLOS ONE: https://journals.plos.org/plosone/s/data-availability.

“If there are ethical or legal restrictions on sharing a sensitive data set, authors should provide the following information within their Data Availability Statement upon submission:

•    Explain the restrictions in detail (e.g., data contain potentially identifying or sensitive patient information)

•    Provide contact information for a data access committee, ethics committee, or other institutional body to which data requests may be sent”

“Please note it is not acceptable for an author to be the sole named individual responsible for ensuring data access.”

If your university has imposed restrictions on sharing of this data, you must provide the contact information for a data access committee (or equivalent institutional committee) that is authorized to review data requests and share data upon approval. One of the authors cannot be named as being responsible for data requests.

Besides this issue, please respond to the remaining comments/questions raised by Reviewer #2.

We look forward to receiving your revised manuscript.

Kind regards,

Jonathan Jacobs

Academic Editor

PLOS ONE

Journal Requirements:

Additional Editor Comments:

The primary remaining issue is data availability. Please see the policy of PLOS ONE: https://journals.plos.org/plosone/s/data-availability.

“If there are ethical or legal restrictions on sharing a sensitive data set, authors should provide the following information within their Data Availability Statement upon submission:

• Explain the restrictions in detail (e.g., data contain potentially identifying or sensitive patient information)

• Provide contact information for a data access committee, ethics committee, or other institutional body to which data requests may be sent”

“Please note it is not acceptable for an author to be the sole named individual responsible for ensuring data access.”

If your university has imposed restrictions on sharing of this data, you must provide the contact information for a data access committee (or equivalent institutional committee) that is authorized to review data requests and share data upon approval. One of the authors cannot be named as being responsible for data requests.

Besides this issue, please respond to the remaining comments/questions raised by Reviewer #2.

Reviewers' comments:

Reviewer's Responses to Questions

**Comments to the Author**

1. If the authors have adequately addressed your comments raised in a previous round of review and you feel that this manuscript is now acceptable for publication, you may indicate that here to bypass the “Comments to the Author” section, enter your conflict of interest statement in the “Confidential to Editor” section, and submit your "Accept" recommendation.

Reviewer #1: All comments have been addressed

Reviewer #2: (No Response)

2. Is the manuscript technically sound, and do the data support the conclusions?

Reviewer #1: Yes

Reviewer #2: Partly

3. Has the statistical analysis been performed appropriately and rigorously? 

Reviewer #1: Yes

Reviewer #2: Yes

4. Have the authors made all data underlying the findings in their manuscript fully available?

Reviewer #1: No

Reviewer #2: No

5. Is the manuscript presented in an intelligible fashion and written in standard English?

Reviewer #1: Yes

Reviewer #2: Yes

6. Review Comments to the Author

Reviewer #1: All the comments and concerns have been satisfactorily addressed except for the data availability issue.

Reviewer #2: I appreciate the authors’ time and effort to strengthen this manuscript. Most reviewers’ comments have been addressed. Below are the remaining concerns/questions.

1. Data Sharing (Major concern): The authors have not followed the PLOS Data policy which requires authors to make all data underlying the findings described in their manuscript fully available without restriction. The authors cited “potentially sensitive patient information” as the rationale for not making the data available. It is unclear how deidentified patients’ data contain any potentially sensitive patient information. It is also questionable whether a data access request from a fellow researcher will be honored in the future.

Introduction:

2. The added hypothesis was “dysbiosis of the microbiota is associated with pathogenesis of PMDs.” Please define “dysbiosis.” Please stay away from causal language as this small cross-sectional study could not establish causation.

3. The authors used the term “low grade inflammation.” What does it mean? What are the characteristics/markers of low grade inflammation?

Methods:

4. I am puzzled by the fact that 7 symptom-free women reported “symptom interference with social life”. Please explain.

5. Some details on sample collection are missing. What is the length of time from collection to receipt by the lab? Participant compliance can be an issue when you ask participants to store stool samples in 4 Celsius degrees (in their fridge??). There can be issues related to sample integrity. Please comment how you approached participant compliance or acknowledge this as a limitation.

6. Please describe the rationale to assess bacteria translocation indirectly rather than directly.

7. Those who took drospirenone-containing oral contraceptives were excluded from the study. What about other types of hormonal contraceptives? Did you collect the data on those and consider them when interpreting the findings?

Results

8. It is unclear why LEfSE plots could not be used.

9. Can you report the effect size between groups for gut microbiome data?

Discussion

10. Please explain the negative findings regarding alpha diversity.

Overall

11. The manuscript can benefit from proof-reading by an English editor.

7. PLOS authors have the option to publish the peer review history of their article (what does this mean?). If published, this will include your full peer review and any attached files.

Reviewer #1: **Yes: **Swapna Mahurkar-Joshi

Reviewer #2: No

---

## [Author Response · Author response to Decision Letter 1]

24 Mar 2022

Professor Emily Chenette

Editor-in-Chief 

PLOS ONE

March 24, 2022

Dear Professor Chenette,

Thank you for reconsidering our manuscript entitled “Characteristics of the gut microbiota in women with premenstrual symptoms: A cross-sectional study” as an Original Research article for publication in PLOS ONE. We have revised our manuscript in response to the many helpful comments of the reviewers, as shown below. 

Additional Editor Comments:

The primary remaining issue is data availability. Please see the policy of PLOS ONE: https://journals.plos.org/plosone/s/data-availability.

“If there are ethical or legal restrictions on sharing a sensitive data set, authors should provide the following information within their Data Availability Statement upon submission:

• Explain the restrictions in detail (e.g., data contain potentially identifying or sensitive patient information)

• Provide contact information for a data access committee, ethics committee, or other institutional body to which data requests may be sent”

“Please note it is not acceptable for an author to be the sole named individual responsible for ensuring data access.”

If your university has imposed restrictions on sharing of this data, you must provide the contact information for a data access committee (or equivalent institutional

committee) that is authorized to review data requests and share dataupon approval. One of the authors cannot be named as being responsible for data requests.

Re: According to this comment, we have added the relevant information as supporting data in files S1 to S4. 

Reviewer: 1

All the comments and concerns have been satisfactorily addressed except for the data availability issue.

Re: Thank you for appreciating our efforts. We have added the requested information as supporting data in files S1 to S4. 

Reviewer: 2

1. Data Sharing (Major concern): The authors have not followed the PLOS Data policy which requires authors to make all data underlying the findings described in their manuscript fully available without restriction. The authors cited “potentially sensitive patient information” as the rationale for not making the data available. It is unclear how deidentified patients’ data contain any potentially sensitive patient information. It is also questionable whether a data access request from a fellow researcher will be honored in the future.

Re: Initially, we thought that the data could not be made public because they were clinical data. However, as you point out, the data are completely anonymized and thus not individuals are not identifiable. Therefore, we have made the data available by including it as supporting information in files S1 to S4. 

Introduction

2. The added hypothesis was “dysbiosis of the microbiota is associated with pathogenesis of PMDs.” Please define “dysbiosis.” Please stay away from causal language as this small cross-sectional study could not establish causation.

Re: As stated in the limitations, we understand that it is not possible to state causality based on the results of this study. We have revised the manuscript to avoid any causal language. The purpose of our study was to explore the relationship between the gut microbiota and PMDs in an exploratory manner. To make this clear, we have revised the text on page 5, lines 87–89.

3. The authors used the term “low grade inflammation.” What does it mean? What are the characteristics/markers of low grade inflammation?

Re: In reference 22, low grade inflammation is defined as a CRP level of greater than 3 mg/L. We have added this information on page 5, lines 75–76.

Methods:

4. I am puzzled by the fact that 7 symptom-free women reported “symptom interference with social life”. Please

explain.

Re: We apologize for the lack of clarity in our explanation. As we describe in the Introduction on page 4, line 56, the prevalence of premenstrual symptoms is high (80%–90%), so completely symptom-free women are rare. The inclusion criteria for the N group included no symptoms listed in the PSQ showing a moderate or higher level, as described on page 6, lines 112–113, so this means that the N group includes those with mild premenstrual symptoms. These women are not completely free of premenstrual symptoms. The seven patients excluded from the N group had multiple mild premenstrual symptoms and mild interference of social life due to these symptoms as assessed by the PSQ. To clarify, we have revised the text on page 7, lines 124–128. In addition, we changed the wording of the description of the N group in Fig. 1 from “Not reporting premenstrual symptoms” to “Not reporting serious premenstrual symptoms”. For consistency, the text on page 2, line 26, 28 and on page 6, line 105, was also changed accordingly.

5. Some details on sample collection are missing. What is the length of time from collection to receipt by the lab? Participant compliance can be an issue when you ask participants to store stool samples in 4 Celsius degrees (in their fridge??). There can be issues related to sample integrity. Please comment how you approached participant compliance or acknowledge this as a limitation.

Re: In this study, special sampling tubes were used for DNA collection from stool specimens. Within these tubes was a storage solution containing guanidine, which is guaranteed to be stable for one month at room temperature or 4 °C; therefore, subjects did not need to store their specimens in a refrigerator. We have added this information on page 9, lines 177–182. 

6. Please describe the rationale to assess bacteria translocation indirectly rather than directly.

Re: Analyzing the DNA of intestinal bacteria in blood is more labor-intensive and costly than indirect marker measurements. Given the exploratory nature of this study, we chose to examine bacterial translocation via an indirect method. We have added this explanation on page 18, lines 342–344. 

7. Those who took drospirenone-containing oral contraceptives were excluded from the study. What about other types of hormonal contraceptives? Did you collect the data on those and consider them when interpreting the findings?

Re: Drospirenone-containing OCs were excluded because, unlike other conventional OCs, they have a proven therapeutic effect on PMDs. The percentage of Japanese women who use OCs as a contraceptive method is low (0.9%, as reported by the WHO in 2015), and as a result, none of the subjects in this study were OC users. We have added additional information to explain this on page 6, lines 110–111 and 117.

 Results

8. It is unclear why LEfSE plots could not be used.

Re: In the first revision, we seem to have misunderstood the reviewer's comments on “LEfSE plots”. Assuming that "LEfSE plots" refers to cladograms, we have added these data as a new Fig 4A. We also changed the figure legend for Fig 4 accordingly on page 15, lines 300–302 and page 16, lines 303–306. 

. 

 9. Can you report the effect size between groups for gut microbiome data?

Re: We analyzed the effect sizes and have added these data to Fig 3.

Discussion

10. Please explain the negative findings regarding alpha diversity.

Re: Alpha diversity was already described in the original version of the study in the case of MDD and inflammatory bowel disease, but a comparison with the PMD results obtained here was missing. Therefore, an explanation of alpha diversity relating to PMDs and MDD was added on page 18, line 354. 

Overall

11. The manuscript can benefit from proof-reading by an English

editor.

Re: Our revised manuscript has been reviewed by a native English-speaking expert editor.

We hope our revised manuscript is now acceptable for publication in PLOS ONE. We look forward to hearing from you at your earliest convenience.

Yours sincerely,

Takashi Takeda, MD, PhD

---

## [Editor Report · Decision Letter 2]

4 Apr 2022

PONE-D-21-30216R2Characteristic of the gut microbiota in women with premenstrual symptoms : a cross-sectional studyPLOS ONE

Dear Dr. Takeda,

Thank you for submitting your manuscript to PLOS ONE. After careful consideration, we feel that it has merit but does not fully meet PLOS ONE’s publication criteria as it currently stands. Therefore, we invite you to submit a revised version of the manuscript that addresses the points raised during the review process.

The comments on the prior revision have been adequately addressed, with the exception that the data made available consists of derived datasets (genus and phylum level count table) rather than raw sequence data as is standard in the field (i.e. fastq files). Could the authors deposit the raw sequence data in a public repository (e.g. NCBI SRA) or provide a reason why the raw sequence data are no longer available?

We look forward to receiving your revised manuscript.

Kind regards,

Jonathan Jacobs

Academic Editor

PLOS ONE

Journal Requirements:

Additional Editor Comments (if provided):

The authors have adequately addressed the comments on the prior revision of their manuscript, with the exception that the data made available consists of derived datasets (genus and phylum level count table) rather than raw sequence data as is standard in the field (i.e. fastq files). Could the authors deposit the raw sequence data in a public repository (e.g. NCBI SRA) or provide a reason why the raw sequence data are no longer available?
---

## [Author Response · Author response to Decision Letter 2]

28 Apr 2022

Professor Emily Chenette

Editor-in-Chief 

PLOS ONE

April 28, 2022

Dear Professor Chenette,

Thank you for reconsidering our manuscript entitled “Characteristics of the gut microbiota in women with premenstrual symptoms: A cross-sectional study” as an Original Research article for publication in PLOS ONE. We have revised our manuscript in response to the helpful comments of the editor, as shown below. 

Additional Editor Comments:

The authors have adequately addressed the comments on the prior revision of their manuscript, with the exception that the data made available consists of derived datasets (genus and phylum level count table) rather than raw sequence data as is standard in the field (i.e. fastq files). Could the authors deposit the raw sequence data in a public repository (e.g. NCBI SRA) or provide a reason why the raw sequence data are no longer available?

Re: Following this comment, we registered the raw sequence data to the public repository (DRA) and added the accession numbers on page 23, line 442–445.

In this revision, we reviewed the description of the method section and found some mistakes. Therefore, we have revised it as described on page 10, line 202–204.

We hope our revised manuscript is now acceptable for publication in PLOS ONE. We look forward to hearing from you at your earliest convenience.

Yours sincerely,

Takashi Takeda, MD, PhD

---

## [Editor Report · Decision Letter 3]

2 May 2022

Characteristic of the gut microbiota in women with premenstrual symptoms : a cross-sectional study

PONE-D-21-30216R3

Dear Dr. Takeda,

We’re pleased to inform you that your manuscript has been judged scientifically suitable for publication and will be formally accepted for publication once it meets all outstanding technical requirements.

Kind regards,

Jonathan Jacobs

Academic Editor

PLOS ONE
---

## [Editor Report · Acceptance letter]

17 May 2022

PONE-D-21-30216R3 

Characteristics of the gut microbiota in women with premenstrual symptoms: a cross-sectional study 

Dear Dr. Takeda:

I'm pleased to inform you that your manuscript has been deemed suitable for publication in PLOS ONE. Congratulations! Your manuscript is now with our production department. 

Kind regards, 

on behalf of

Dr. Jonathan Jacobs 

Academic Editor

PLOS ONE